# Nutritional Adequacy and Latent Tuberculosis Infection in End-Stage Renal Disease Patients

**DOI:** 10.3390/nu11102299

**Published:** 2019-09-26

**Authors:** Seung Don Baek, Soomin Jeung, Jae-Young Kang

**Affiliations:** 1Division of Nephrology, Department of Internal Medicine, Mediplex Sejong Hospital, Incheon KS006, Korea; soominj@sejongh.co.kr; 2Division of Nephrology, Department of Internal Medicine, Sejong General Hospital, Bucheon KS009, Korea; jykangmd@naver.com

**Keywords:** dialysis, latent tuberculosis, kidney failure

## Abstract

Background: Latent tuberculosis infection (LTBI) is prevalent in end-stage renal disease (ESRD) patients. The risk of tuberculosis activation is also high. The appropriate LTBI screening and treatment is required in this population. Meanwhile, whether hemodialysis adequacy is associated with LTBI in the ESRD population is unclear. In this study, we aimed to investigate the association between hemodialysis adequacy and LTBI in ESRD patients. Methods: In the present cross-sectional study, we reviewed all outpatient-based ESRD patients in our artificial kidney room. Interferon gamma release assay (IGRA) was used for the diagnosis of LTBI. Clinical variables including nutritional adequacy (i.e., normalized protein catabolic rate, nPCR) and dialysis adequacy (i.e., Kt/V) were compared between IGRA-positive and IGRA-negative patients. Results: A total of 90 patients were enrolled, of which 20 (22.2%) had positive IGRA results using the QuantiFERON-TB method. Old fibrotic changes and nPCR (g/kg/day) were significantly different between IGRA-positive and IGRA-negative patients (both *p* < 0.005), while serum albumin and Kt/V were comparable (*p* = 0.429 and *p* = 0.590, respectively). Normalized PCR remained to be significant in a multivariate logistic regression analysis (adjusted hazard ratio, 0.911 (0.861–0.963); *p* = 0.001). The cutoff nPCR value less than 0.87 g/kg/day had an adjusted hazard ratio of 7.74 (1.77–33.74) for predicting LTBI. Patients with nPCR value less than 0.87 g/kg/day were older and had lower serum hemoglobin, albumin, calcium concentration, and Kt/V levels than those with nPCR value greater than 0.87 g/kg/day. Conclusions: Nutritional adequacy, especially when assessing nPCR value, was associated with LTBI, while dialysis adequacy was not associated with LTBI.

## 1. Introduction

Latent tuberculosis infection (LTBI) is prevalent in South Korea, a country with an intermediate tuberculosis burden [1]. LTBI treatment has been emphasized in patients who are immunocompromised or dialyzed. Among those, end-stage renal disease (ESRD) patients are highly susceptible to LTBI and are also at high risk of developing active disease [2]. Therefore, appropriate and optimal screening for LTBI is required in dialysis population.

Protein-energy malnutrition has a significant impact on patient outcomes. Protein intake estimated by normalized protein catabolic rate (nPCR) and dialysis adequacy expressed by Kt/V were considered the major determinants of nutritional status in the ESRD population [3]. Although malnutrition is a risk factor of tuberculosis (TB) infection and both could interact with each other, association between hemodialysis adequacy and risk of LTBI has not been studied yet.

In the present study, we performed interferon gamma release assay (IGRA) using QuantiFERON-TB for the diagnosis of LTBI. We investigated several clinical and nutritional variables including demographics, laboratory values, nPCR, and Kt/V to determine the association with LTBI. Thereafter, we aimed to find the independent risk factors associated with LTBI and to identify patients at risk of active TB.

## 2. Materials and Methods

### 2.1. Study Population 

In this cross-sectional study, we included all outpatient-based hemodialysis patients in Mediplex Sejong Hospital from March 2017 to August 2019. We routinely performed IGRA test in all ESRD patients who underwent their first dialysis at our center or were referred from other hospitals. During the study, a total of 90 patients who had chronic dialysis for at least 3 months and underwent IGRA test were finally enrolled. The study was approved by the local ethics committee (2019-088). Informed consent was waived due to the retrospective nature of this study.

### 2.2. Dialysis Prescription

Patients underwent a 3.5–4-h hemodialysis session twice to thrice per week, with blood flow rate ranging from 200 mL/min to 350 mL/min using a bicarbonate buffered dialysate and biocompatible membranes. There were six peritoneal dialysis patients, of whom two underwent automated and four underwent continuous ambulatory peritoneal dialysis. 

### 2.3. Data Collection

Baseline demographic and clinical data were obtained, including age, gender, body mass index (BMI), dialysis type, dialysis vintage, comorbidity, blood pressure, heart rate, and interdialytic weight gain. Comorbidities included diabetes, hypertension, chronic glomerulonephritis, coronary artery disease, and cerebrovascular disease. History of prior TB treatment and administration of immunosuppressant medication were evaluated and chest radiographs were reviewed. Laboratory values included serum levels of hemoglobin, albumin, ferritin, transferrin saturation, calcium, phosphorus, and intact parathyroid hormone. All clinical parameters were collected when IGRA test was being performed.

### 2.4. Hemodialysis Adequacy

Single-pool Kt/V [4] and nPCR were recorded. Normalized PCR for dry body mass was calculated from the urea generation rate [5]. We did not consider residual renal function in this study. Hemodialysis adequacy was repeated every 3 months. The formula is as follows:Kt/V = −ln (R − 0.008 × t) + (4 − 3.5R) × UF/W,
where R is the post-dialysis/pre-dialysis blood urea nitrogen, t is the dialysis hours, UF is the pre- and post-dialysis weight change, and W is the post-dialysis weight.
Normalized PCR = PCR/standard weight (g/kg/day).
PCR = 0.22 + 0.036 × (next pre-dialysis blood urea nitrogen − post-dialysis blood urea nitrogen) × 24/dialysis interval.
Standard weight = total body water/0.58.

In peritoneal dialysis patients,
PCR = 6.25 × (urea appearance + 1.81 + [0.031 × lean body weight]).
Urea appearance (g/day) = volume of dialysate × Urea concentration of dialysate.

### 2.5. Interferon Gamma Release Assay

IGRA was performed according to the manufacturer’s instructions. Any interferon gamma response greater than 0.35 IU/mL using an enzyme-linked immunoassay QuantiFERON-TB Gold was considered positive.

### 2.6. Statistical Analysis

All data were expressed as mean ± standard deviation or median and interquartile ranges according to the distribution. Group difference was analyzed using independent Student’s t-test or Mann-Whitney U test. All parameters were entered as covariates in a univariate logistic regression analysis. A multivariate analysis using identified factors that showed a statistical significance in a univariate analysis (*p* < 0.05) was performed to determine the independency. Adjusted hazard ratio (HR) was calculated. A receiver operating characteristic (ROC) curve revealed the best cutoff for prediction using Youden’s index. *p* < 0.05 was considered to be statistically significant. Statistical Package for the Social Sciences (SPSS) version 22.0 (SPSS, Chicago, IL, USA) and MedCalc version 14.8.1 (MedCalc, Mariakerke, Belgium) software were used for analysis.

## 3. Results

A total of 90 ESRD patients were included during the study period, with 20 patients (22.2%) showing IGRA positivity. The mean age was 61.6 ± 12.6 years, and 56 patients (62.2%) were male. There were 84 hemodialysis (93.3%) and 6 peritoneal dialysis patients (6.7%). The median dialysis duration was 21 (10–56.5) months. Hemodialysis adequacy showed nPCR of 0.93 ± 0.18 in all patients (*n* = 90) and single-pool Kt/V per session of 1.52 ± 0.24 in hemodialysis patients (*n* = 84).

Between the two groups, old fibrotic lesions and nPCR were significantly different (both *p* < 0.005) (Table 1). Comorbidity, levels of nutritional markers such as serum albumin and ferritin, and mineral bone disease markers such as serum calcium, phosphorus, and intact parathyroid hormone were not significantly different. Dialysis adequacy, that is, Kt/V, was also comparable (*p* = 0.590) (Figure 1).

Next, we determined the independent parameters associated with LTBI. By performing a univariate logistic regression analysis, old fibrotic changes on chest radiograph and nPCR showed a significant association (hazard ratio [HR], 10.83 (3.33–35.14); *p* < 0.005 and HR, 0.911 (0.861–0.963); *p* = 0.001, respectively). Two parameters were still significant in a multivariate analysis (adjusted HR, 8.3 (1.79–38.49); *p* = 0.007 and adjusted HR, 0.912 (0.859–0.969); *p* = 0.003, respectively). 

We determined the best cutoff of nPCR predictive for LTBI using ROC analysis and analyzed in a multivariate analysis again. The ROC analysis revealed that a nPCR cutoff of 0.87 g/kg/day using Youden’s index had a sensitivity of 83.3%, a specificity of 73.1%, and an area under the ROC curve of 0.81 (0.70–0.89). Table 2 shows the two independent factors associated with IGRA positivity and adjusted HR.

We also compared the groups divided by nPCR value greater than 0.87 g/kg/day and less than 0.87 g/kg/day. Patients with nPCR value less than 0.87 g/kg/day were older and had lower serum hemoglobin, albumin, calcium, and Kt/V levels than patients with nPCR value greater than 0.87 g/kg/day (Table 3). There was a non-significant trend toward more peritoneal dialysis and hypertension patients in the group with nPCR less than 0.87 g/kg/day (*p* = 0.058 and *p* = 0.089, respectively).

## 4. Discussion

In the present study, we found that the prevalence of LTBI in ESRD patients using the IGRA test was 22.2%, which was comparable to that in the previous report [6]. Regarding hemodialysis adequacy, nPCR was significantly associated with LTBI, while other nutritional markers such as serum albumin, ferritin, and dialysis adequacy were not associated with LTBI. We showed that low protein intake was associated with LTBI risk in the ESRD population. 

LTBI is prevalent in Korea, with a prevalence of 33% [7]. The treatments and interventions performed to prevent LTBI decreased its prevalence, incidence, and mortality rate over the past few decades [7]. However, the prevalence of LTBI in ESRD patients is significantly high. It is up to 41.9% in one Korean study, which is higher than that of our study population [8]. In this context, screening dialysis patients for LTBI and performing preventive treatment are emphasized to reduce the transmission. So far, T-cell-based assays and tuberculin skin test were considered as screening methods, with the latter having a high rate of cutaneous anergy and low specificity. Among T-cell-based assays, QuantiFERON-TB Gold in tubes was better than tuberculin skin test for detecting LTBI in the ESRD population [6]. QuantiFERON-TB Gold in tubes was a more accurate method even compared to T-SPOT test [9].

Malnutrition has a significant effect on morbidity and mortality in dialysis patients. Malnutrition, inflammation, and atherosclerosis commonly coexist and interact in chronic kidney disease patients [10]. Levels of nutritional markers, especially serum albumin and prealbumin, were predictive of death in ESRD patients [11]. Protein intake shown as nPCR combined with serum albumin were predictive of mortality in chronic hemodialysis (HD) patients [12]. In our study, we found that lower nPCR was associated with LTBI. We showed that a nPCR value less than 0.87 g/kg/day showed a 7-fold higher risk of LTBI compared to a nPCR value greater than 0.87 g/kg/day. The K/DOQI guidelines suggested that nPCR values between 1.0 and 1.2 g/kg/day and nPCR values less than 0.8 g/kg/day or greater than 1.4 g/kg/day were significantly associated with mortality. Unfortunately, we could not evaluate the exact causes of malnutrition. In addition, fibrotic changes on chest radiography were significant predictors. We excluded the possibility of active TB in all LTBI patients, as demonstrated by a lack of general and respiratory symptoms, and pulmonary changes in serial chest radiography.

Meanwhile, serum albumin is a more potent predictor for survival, even reflecting disease severity rather than malnutrition [13]. In a large cohort study, survival was determined by nutritional status as reflected by the albumin and prealbumin levels, not by hemodialysis adequacy such as nPCR and Kt/V [14]. Compared to serum albumin as a nutritional marker, the nPCR had several limitations. It might be underestimated by malnourishment or overestimated by high protein intake and urea rebound. Residual renal function, especially in peritoneal dialysis patients, also contributes to the underestimation. As a comprehensive approach to evaluate nutritional status, a panel of nutritional indices rather than single measurement should be considered in these populations. Unfortunately, an association between serum albumin and LTBI was not observed in our study. Dialysis adequacy also was not associated with LTBI. There are some differences in that nPCR is an estimation related to protein consumption, catabolism, and dialytic removal, while serum albumin is a blood protein related to general nutrition, inflammation, and is removed minimally by dialysis. Considering the nature of the study, a large longitudinal study regarding the association between various nutritional markers and LTBI or active TB is required.

The association between malnutrition and LTBI needs further evaluation. LTBI subjects with low BMI were at risk of developing active TB [15]. In dialysis patients who already had a high risk of prevalent LTBI, malnutrition is an important risk factor for active tuberculosis, partly because these patients have decreased cell-mediated immunity [16]. Conversely, TB patients have significantly decreased body weight with loss of both lean and fat body mass [17]. Cytokine activation and abnormal protein metabolism were suggested to bidirectionally interact with TB and malnutrition [18]. Although nPCR was one of the surrogate markers associated with nutritional status and hemodialysis adequacy, we revealed that nPCR was associated with LTBI, and lower nPCR value was significantly associated with LTBI. We suggest a regular monitoring of nPCR might give valuable information beyond hemodialysis adequacy in ESRD patients. 

This study has several limitations. First, a small retrospective design could not make a definite causal relationship. Second, as we pointed out in the discussion, the ability of nPCR to predict nutritional status has several limitations. We believe our results should be interpreted with consideration of the limitations of nPCR. Third, although IGRA is considered to be better than other diagnostic tools such as the tuberculin skin test, it is also not a gold standard test for LTBI detection. Spontaneous conversion and reversion within subjects has been reported [19].

## 5. Conclusions

We showed that nPCR was associated with LTBI in ESRD patients. Although patients with lower nPCR value had lower serum albumin, albumin level was not significantly associated with LTBI. The association between low protein intake by prescribed dietary restrictions and inadequate monitoring [20] and LTBI was suggested. Nutritional interventions to prevent protein-energy malnutrition might reduce the risk of LTBI. Further longitudinal studies are required to determine the risk of malnutrition for LTBI or the risk of LTBI for malnutrition in dialysis patients.

## Figures and Tables

**Figure 1 nutrients-11-02299-f001:**
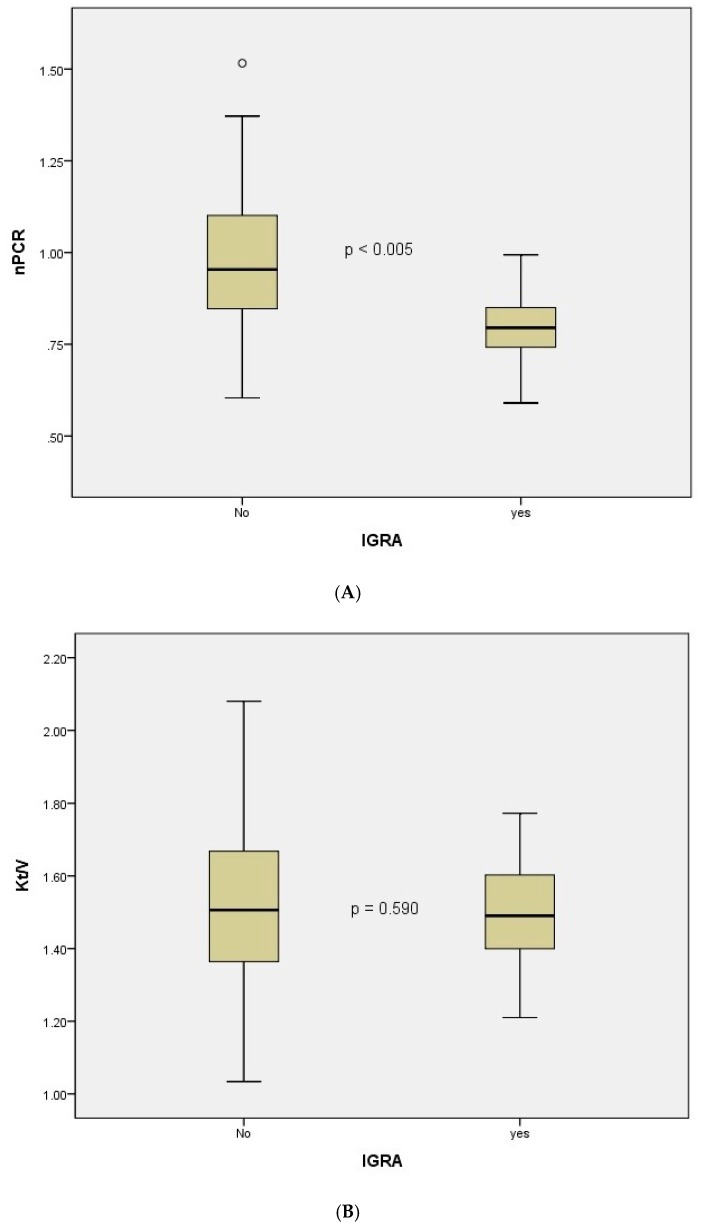
Box plot comparison of normalized protein catabolic rate (**A**) and Kt/V (**B**) between the two groups.

**Table 1 nutrients-11-02299-t001:** Baseline characteristics of the groups according to interferon gamma release assay.

	Negative IGRA (*N* = 70)	Positive IGRA (*N* = 20)	*p* Value
Age (years)	61.3 ± 13.5	62.8 ± 9.1	0.656
Male	44 (62.9%)	12 (60.0%)	1.000
Body mass index (kg/m^2^)	23.0 ± 3.1	23.5 ± 3.5	0.525
Dialysis type (hemodialysis vs. peritoneal dialysis)	64 (91.4%) vs. 6 (8.6%)	20 (100.0%) vs. 0 (0%)	0.397
Dialysis vintage (months)	18.0 (10.0–53.0)	23.0 (15.0–60.0)	0.302
Comorbidity			
Diabetes	33 (47.8%)	13 (65.0%)	0.272
Hypertension	65 (94.2%)	20 (100.0%)	0.625
Chronic glomerulonephritis	13 (18.8%)	1 (5.0%)	0.251
Coronary artery disease	28 (40.6%)	6 (30.0%)	0.551
Cerebrovascular disease	3 (4.3%)	3 (15.0%)	0.243
Prior tuberculosis treatment	1 (1.4%)	1 (5.0%)	0.931
Old fibrotic changes on chest radiograph	7 (10.1%)	11 (55.0%)	<0.005
Immunosuppressant medication	3 (4.3%)	0 (0.0%)	0.806
Systolic blood pressure (mmHg)	144.8 ± 20.1	147.0 ± 22.4	0.679
Diastolic blood pressure (mmHg)	69.7 ± 14.9	66.0 ± 14.7	0.334
Heart rate (rate/min)	69.5 (63.0–77.0)	72.5 (68.5–78.5)	0.158
Interdialytic weight gain (hemodialysis patients only)	2.4 ± 1.0	2.4 ± 1.1	0.919
Hemoglobin, g/dL	10.1 (9.5–10.8)	10.2 (9.6–10.6)	0.829
Albumin, g/dL	3.8 (3.5–4.0)	3.8 (3.4–3.9)	0.429
Ferritin, ug/L	201.0 (98.2–309.9)	204.6 (115.4–296.4)	0.884
Transferrin saturation, %	26.7 (21.3–36.0)	32.3 (25.3–38.5)	0.367
Calcium, mg/dL	8.9 ± 0.7	8.9 ± 0.7	0.957
Phosphorus, mg/dL	4.4 (3.7–5.5)	4.4 (3.8–5.2)	0.775
Intact parathyroid hormone, pg/mL	140.0 (87.7–238.0)	194.0 (85.3–321.0)	0.338
Single-pool Kt/V (hemodialysis patients only, per dialysis session)	1.5 ± 0.3	1.5 ± 0.2	0.590
Normalized protein catabolic rate, g/kg/day	1.0 ± 0.2	0.8 ± 0.1	<0.005

IGRA, interferon gamma release assay.

**Table 2 nutrients-11-02299-t002:** Variables associated with latent tuberculosis infection.

Variables	Adjusted Hazard Ratio (95% Confidence Interval)	*p* Value
Old fibrotic changes	11.93 (3.17–44.85)	<0.005
nPCR less than 0.87 g/kg/day	7.74 (1.77–33.74)	0.006

nPCR, normalized protein catabolic rate.

**Table 3 nutrients-11-02299-t003:** Baseline characteristics of the groups according to normalized protein catabolic rate.

Normalized Protein Catabolic Rate (g/kg/day)	>0.87 (*N* = 41)	≤0.87 (*N* = 49)	*p* Value
Age (years)	58.5 ± 12.0	64.2 ± 12.6	0.031
Male	24 (58.5%)	32 (65.3%)	0.659
Body mass index (kg/m^2^)	22.8 ± 3.1	23.3 ± 3.3	0.445
Dialysis type (hemodialysis vs. peritoneal dialysis)	41 (100.0%) vs. 0 (0.0%)	43 (87.8%) vs. 6 (12.2%)	0.058
Dialysis vintage (months)	23.0 (12.0–60.0)	20.0 (9.0–44.5)	0.358
Comorbidity			
Diabetes	19 (46.3%)	27 (56.2%)	0.472
Hypertension	37 (90.2%)	48 (100.0%)	0.089
Chronic glomerulonephritis	9 (22.0%)	5 (10.4%)	0.231
Coronary artery disease	12 (29.3%)	22 (45.8%)	0.166
Cerebrovascular disease	1 (2.4%)	5 (10.4%)	0.284
Prior tuberculosis treatment	0 (0.0%)	2 (4.2%)	0.545
Old fibrotic changes on chest radiograph	6 (14.6%)	12 (25.0%)	0.343
Immunosuppressant medication	2 (4.9%)	1 (2.1%)	0.889
Pre-dialysis systolic blood pressure (mmHg)	148.8 ± 22.7	142.4 ± 18.3	0.141
Pre-dialysis diastolic blood pressure (mmHg)	71.4 ± 12.1	66.7 ± 16.6	0.127
Pre-dialysis heart rate (rate/min)	69.0 (64.0–75.0)	72.0 (64.0–80.0)	0.200
Interdialytic weight gain (hemodialysis patients only)	2.4 ± 1.0	2.4 ± 1.1	0.901
Hemoglobin, g/dL	10.3 (9.9–0.8)	9.9 (9.3–10.5)	0.038
Albumin, g/dL	3.9 (3.7–4.0)	3.7 (3.4–3.9)	0.003
Ferritin, ug/L	179.8 (95.0–308.5)	221.5 (134.7–309.9)	0.509
Transferrin saturation, %	28.6 (21.1–39.1)	28.0 (23.2–36.0)	0.946
Calcium, mg/dL	9.1 ± 0.6	8.7 ± 0.7	0.010
Phosphorus, mg/dL	4.5 (3.8–5.5)	4.3 (3.7–5.3)	0.760
Intact parathyroid hormone, pg/mL	153.0 (97.0–239.0)	141.0 (71.1–243.0)	0.840
Single-pool Kt/V (hemodialysis patients only, per dialysis session)	1.6 ± 0.2	1.4 ± 0.2	0.026

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
