# Peer review of "Nutritional Adequacy and Latent Tuberculosis Infection in End-Stage Renal Disease Patients"

_nutrients, 2019, doi:10.3390/nu11102299_

Round 1
Reviewer 1 Report
Line 32: Better to take out the word ''especially''. I do not think you mean that treating all LTBI has been emphasized. Did you? if so you need to give us reference for that.
Line 51: Should be for at least 3 months.
Line 97: Here it is better to add the number of patients who had IGRA indeterminant values. If there was none please state that as well.
Table 1: It is not clear where the patients with peritoneal dialysis lie, are they in IGRA positive or IGRA negative group? If you do not want to put that in the table you can put that information somewhere in the text.
Line123:I do not think you need the words ''is present'' in that sentence
Line 130: replace a group with the group
Line 140: the statement in line 140 ''treatment and intervention... of LTBI...decades'' needs reference
Line 142: but in this study the LTBI is lower than the population prevalence, explain why. Could it be the sample size?
Line 156: Why did you not follow the K/DOQ guideline and used nPCR values<0.8g rather than using 0.87g. Was it because of significant result? If not would it not be better to use a cutoff value which was shown to be associated with mortality? You can also show both.
Line 173: were instead of was
Other comment:
Your other significant result was association of old fibrotic changes with LTBI, but this finding was not discussed in the discussion part. Sometimes this fibrotic changes could be old TB scars or even active TB. Was active TB excluded in all the LTBI patients? if so how?
Author Response
[Responses to Reviewer 1’s comments]
Reviewer #1:
Line 32: Better to take out the word ''especially''. I do not think you mean that treating all LTBI has been emphasized. Did you? if so you need to give us reference for that.
|
As you recommended, we have removed the word. |
Line 51: Should be for at least 3 months.
|
As you recommended, we have rephrased the sentence. |
Line 97: Here it is better to add the number of patients who had IGRA indeterminant values. If there was none please state that as well.
|
In the Methods section, we defined the positivity of IGRA. In this study, any interferon gamma response greater than 0.35 IU/mL using the enzyme-linked immunoassay QuantiFERON-TB Gold was considered positive; otherwise, it was considered negative. Therefore, none of the patients had indeterminant IGRA values. |
Table 1: It is not clear where the patients with peritoneal dialysis lie, are they in IGRA positive or IGRA negative group? If you do not want to put that in the table you can put that information somewhere in the text.
|
There were 6 peritoneal dialysis patients, all of whom were in the negative IGRA group, as shown in Table 1. |
Line123:I do not think you need the words ''is present'' in that sentence
|
As you recommended, we have removed the words. |
Line 130: replace a group with the group
|
As you recommended, we have replaced those words. |
Line 140: the statement in line 140 ''treatment and intervention... of LTBI...decades'' needs reference
|
As you recommended, we have added a reference for this statement. |
Line 142: but in this study the LTBI is lower than the population prevalence, explain why. Could it be the sample size?
|
As you pointed out, the latent TB prevalence of the population in this study is lower than that of the general population. We believe that the prevalence largely depends on the characteristics and size of the study population. |
Line 156: Why did you not follow the K/DOQ guideline and used nPCR values<0.8g rather than using 0.87g. Was it because of significant result? If not would it not be better to use a cutoff value which was shown to be associated with mortality? You can also show both.
|
The value of 0.87 g was determined by AUC-ROC analysis and Youden’s index, i.e. the best performance. As this was not a longitudinal study, we could not determine the mortality rate. The value of 0.8 g suggested by the K/DOQI guideline also had a significant ability to differentiate the IGRA positivity, for which the AUC-ROC was lower than that of the value of 0.87 g (the AUC-ROC of 0.67 for nPCR 0.8 g vs. the AUC-ROC of 0.81 for nPCR 0.87 g, p < 0.05 for direct comparison). |
Line 173: were instead of was
|
As you pointed out, we have corrected the word. |
Other comment:
Your other significant result was association of old fibrotic changes with LTBI, but this finding was not discussed in the discussion part. Sometimes this fibrotic changes could be old TB scars or even active TB. Was active TB excluded in all the LTBI patients? if so how?
|
As you pointed out, we have added the following sentence in Line 157: “In addition, fibrotic changes on chest radiography were significant predictors. We excluded the possibility of active TB which showed old fibrotic changes and IGRA positivity, as demonstrated by a lack of general, respiratory symptoms, and pulmonary changes in serial chest radiography.” |
Reviewer 2 Report
This is quite interesting study about the prevalence of latent tuberculosis in dialysed patients and its correlation with patients nutritional condition. It was reported that ESRD patients have 6-25-fold higher TB incidence rates, and mortality during treatment is 2-3-fold higher (Epidemiology, detection, and management of tuberculosis among end-stage renal disease patients. Okada RC, Barry PM, Skarbinski J, Chitnis AS.Infect Control Hosp Epidemiol. 2018 Nov;39(11):1367-1374. doi: 10.1017/ice.2018.219), so screening for latent tuberculosis is of special importance. Interestingly, Authors show that the risk of latent tuberculosis is highly dependent on patient's nutritional status, but not on hemodialysis adequacy.
However:
1) Still there is no adequate test in diagnosing latent tuberculosis, in fact researchers showed large discrepancy between the tests (Discordance in latent tuberculosis (TB) test results in patients with end-stage renal disease.Southern J, Sridhar S, Tsou CY, Hopkins S, Collier S, Nikolayevskyy V, Lozewicz S, Lalvani A, Abubakar I, Lipman M.Public Health. 2019 Jan;166:34-39. doi: 10.1016/j.puhe.2018.09.023.). Also IGRA test are no ideal, did Authors think about other test to diagnose latent TB? Patients may have false negative results. This may explain low LTBI prevalence in this study.
2) Did Authors try to analyze causes of patients malnutrition? Maybe due to their nonadherence to diet in kidney failure? Any addictions? The quality of kidney replacement therapy seems to not have any significance, however type = peritoneal dialysis.
3) Were patients tested with IGRA test previously? Any conversion was seen? fibrotic changes (old?) in the X-rays were found and they could be observed for longer time.
4) small language corrections are needed (line 25 "calcium centration").
Author Response
[Responses to the Reviewer 2’s comments]
1) Still there is no adequate test in diagnosing latent tuberculosis, in fact researchers showed large discrepancy between the tests (Discordance in latent tuberculosis (TB) test results in patients with end-stage renal disease.Southern J, Sridhar S, Tsou CY, Hopkins S, Collier S, Nikolayevskyy V, Lozewicz S, Lalvani A, Abubakar I, Lipman M.Public Health. 2019 Jan;166:34-39. doi: 10.1016/j.puhe.2018.09.023.). Also IGRA test are no ideal, did Authors think about other test to diagnose latent TB? Patients may have false negative results. This may explain low LTBI prevalence in this study.
|
As you pointed out, and as we discussed in our Limitations section, IGRA is not a gold standard test for detecting latent TB infection. So far, however, T-cell based assays shows better specificity than other conventional tests such as the tuberculin skin test (Ann Thorac Med. 2009 Jan-Mar; 4(1): 5–9). We agreed that there was a possibility of false negativity of latent TB infection diagnosis in our study. |
2) Did Authors try to analyze causes of patients malnutrition? Maybe due to their nonadherence to diet in kidney failure? Any addictions? The quality of kidney replacement therapy seems to not have any significance, however type = peritoneal dialysis.
|
As you pointed out, we have added the following sentence in Line 157: Unfortunately, we could not evaluate the exact causes of malnutrition. The patients with peritoneal dialysis showed lower nPCR compared with the patients with hemodialysis, albeit with no statistical significance (p = 0.058). As we did not consider residual renal function, the nPCR values of peritoneal dialysis patients might have been underestimated. Poor appetite, nutritional loss via dialysis, and chronic uremia and inflammation were suggested as possible mechanisms (Protein Nutrition and Malnutrition in CKD and ESRD, Nutrients 2017, 9, 208; d). |
3) Were patients tested with IGRA test previously? Any conversion was seen? fibrotic changes (old?) in the X-rays were found and they could be observed for longer time.
|
In this study, since all patients were naïve to the IGRA test, conversion could not be determined. |
4) small language corrections are needed (line 25 "calcium centration").
|
As you pointed out, we have corrected the word. |